# Burnout Among Hospital Nurses in Kazakhstan

**DOI:** 10.3390/nursrep15030092

**Published:** 2025-03-06

**Authors:** Makhigul Maxudova, Dinara Ospanova, Areti Stavropoulou, Lyazzat Alibekova, Gulnar Sultanova, Galina Veklenko, Kundyzay Tobzhanova

**Affiliations:** 1Department of Nursing, Al-Farabi Kazakh National University, Almaty 050040, Kazakhstan; maksudova63@gmail.com (M.M.); dinara.ospanova.alm@gmail.com (D.O.); 2Department of Public Health and Health Care, M. Ospanov West Kazakhstan Medical University, Aktobe 030019, Kazakhstan; 3Department of Nursing, University of West Attica, 122 43 Athens, Greece; astavropoulou@uniwa.gr; 4High Medical College, Public Health Department of Almaty, Almaty 050020, Kazakhstan; 5Department of Propaedeutics of Internal Diseases, M. Ospanov West Kazakhstan Medical University, Aktobe 030019, Kazakhstan

**Keywords:** burnout, nurse, MBI-HSS-MP, emotional exhaustion, depersonalization, personal accomplishment

## Abstract

**Background:** Burnout is an important psychological condition that affects the work performance of nurses. It occurs under long-term psychological or emotional stress associated with the work of a nurse. As a result, symptoms such as emotional exhaustion (EE), depersonalization (DP), and reduced personal accomplishment (PA) may be experienced. The purpose of this study was to determine the syndrome of “burnout” using three subscales—emotional exhaustion (EE), depersonalization (DP), and reduced personal accomplishment (PA)—among nurses providing round-the-clock nursing care in multidisciplinary hospitals in the Republic of Kazakhstan. **Methods:** A cross-sectional study was conducted among nurses in the Republic of Kazakhstan working in round-the-clock care. In total, 284 respondents participated in the online survey. The Maslach Burnout Inventory for Medical Personnel (MBI-HSS-MP) was used for collecting data. **Results:** The results demonstrated that burnout occurred among 61.97% of hospital nurses in the Republic of Kazakhstan. EE was identified among 29.23% of the respondents, DP in 60.92% of the respondents, and PA in 38.73%. Burnout, according to these indicators, occurred in 25.7% (EE), 19.37% (DP), and 12.68% (PA) of nurses. The level of burnout was higher among young nursing specialists and more frequent in the southern region of the Republic of Kazakhstan. **Conclusions:** This study revealed a high level of burnout among nurses providing round-the-clock care in multidisciplinary hospital settings. These findings have implications for further research into the impact of burnout on nurses’ work and for developing interventions to reduce potential risks to nursing staff health and improving the quality of nursing care.

## 1. Introduction

The nursing profession involves constant contact with people; therefore, one’s emotional state during such interaction is critical for the provision of quality care [1,2].

Burnout is a psychological condition that is created during long-term emotional stress in the workplace, resulting in the following symptoms: emotional exhaustion, depersonalization, and reduced personal accomplishments [3,4]. This condition among nurses may have negative consequences for patients, as it is associated with a lower quality of care [5]. Factors such as the working conditions and functional responsibilities of nurses can influence its development. The specialty of a nurse’s job is also important [6]. Work associated with an increased level of communication [1], as well as in departments specialized in providing care for incurable patients and those accompanied by constant emotional stress, gradually exhausts the nurse [6,7]. These factors reduce resistance to stress and ultimately lead to emotional exhaustion and increased poor health and reluctance to work [7]. Thus, in the oncology service of Kazakhstan, a high level of emotional exhaustion (EE) was observed in 47% of respondents, a high level of depersonalization (DP) in 63% of respondents, and a high level of reduced personal accomplishment (PA) in 42% of respondents [8].

In the Republic of Kazakhstan, burnout among healthcare workers has been gaining increasing attention from researchers. For instance, in cardiology, the prevalence of emotional exhaustion among nurses was 26% in the pre-COVID-19 period [9].

Burnout leads to nurses leaving employment. In the USA, based on the results of a study conducted in 2018, it was determined that 31.5% (*n* = 418,769) of nurses identified burnout as the reason for leaving their last position [10]. Therefore, identifying burnout among nurses is an important task for providing quality medical care. The purpose of this study was to determine the syndrome of “burnout” using three subscales—emotional exhaustion (EE), depersonalization (DP), and personal accomplishment (PA)—among nurses providing round-the-clock nursing care in multidisciplinary hospitals in the Republic of Kazakhstan.

## 2. Materials and Methods

### 2.1. Study Design and Settings

A cross-sectional study was conducted in November–December 2023 to identify the levels of burnout among nurses providing round-the-clock nursing care in multidisciplinary hospitals in the Republic of Kazakhstan.

This study included eight regional multidisciplinary clinical hospitals located in the following regions of the Republic of Kazakhstan: central, northern, southern, and western. Currently, there are 17 regional multidisciplinary clinical hospitals in the Republic of Kazakhstan. Three hospitals located in these regions, established in 2022, were not included in this study. Accordingly, this study included eight out of fourteen hospitals, which is 57% of the multidisciplinary regional clinical hospitals, in which the total contingent of nurses amounted to 2032 employees.

As this study aimed to study burnout syndrome among nurses providing round-the-clock nursing care in a hospital setting, we included 1056 such specialists working in these hospitals, which is 52% of the total number of nurses.

According to an online calculator for calculating a representative sample for the general population [11], to represent the 1056 nurses providing round-the-clock nursing care in multidisciplinary hospitals, the sample size needed to be 282 (*p* = 0.05). This study covered 14% of the nurses in regional hospitals considered in this study.

### 2.2. Study Participants

This study involved a questionnaire being completed by 284 nurses providing round-the-clock nursing care in multidisciplinary hospitals in the Republic of Kazakhstan. The participants were informed about the purpose of this study and voluntarily took part in it.

Inclusion criteria: at least three years of work experience in their specialty, and working in a regional multidisciplinary hospital.

Exclusion criteria: nurses working in specialized hospitals, including tuberculosis hospitals, psychiatric hospitals, oncology hospitals, etc.

### 2.3. Data Collection

The nurses were asked to complete an online survey. The questionnaire consisted of three main sections designed to collect demographic data for the respondent, the organizational structure of hospitals, and the level of burnout among nurses. The questionnaire used to determine the level of burnout among medical personnel—the MBI-HSS-MP, which was developed based on the three-phase model of C. Maslach and S.E. Jackson [12] and adapted by N. Vodopyanova and E. Starchenkova [13]—consists of 22 statements aimed at the identification of emotional exhaustion, depersonalization, and professional success. The respondents were presented with 22 statements about feelings and experiences related to their work. They were required to read each statement and choose the corresponding occurrence frequency for each statement on a seven-point scale from 0 to 6. If it was never experienced, 0 was to be selected; if it was experienced daily, they were to choose 6 [14]. The results were evaluated based on the summation of the points scored in accordance with the methodology for conducting this survey (Table 1).

Thus, three subscales were calculated for each respondent. The score was recorded for each subscale and distributed among three levels: low, moderate, and high. A high score for EE and DP indicates a high level of burnout, while high scores in PA indicate a lower level of burnout. High burnout is EE ≥ 25, DP ≥ 11, and PA ≤ 30; moderate burnout is EE 16–24, DP 6–10, and PA 36–31; low burnout is EE 0–15, DP 0–5, and PA ≥ 37 [13].

Low levels of the three subscales show a lack of severity of the analyzed indicator of burnout syndrome. A moderate level shows the stage of the formation of a certain indicator of burnout syndrome. A high level confirms that the indicator of burnout syndrome has already formed.

The MBI has high consistency and reliability, which range from 70 to 95% [12]. In this study, the internal consistency (Cronbach’s alpha) of the burnout questionnaire for medical workers (22 questions) was 0.910, and that of the entire questionnaire was 0.840 (33 questions).

### 2.4. Statistical Analysis

The data were described using descriptive and analytical statistics. A correlation analysis was conducted to determine the influence of factors (age, position, and length of service). The statistical package SPSS 23.0 (IBM SPSS Statistics, Chicago, IL, USA) was used to process the data. Significance criteria were calculated at *p* < 0.05.

### 2.5. Ethical Consideration

An information letter about the purpose and objectives of the study was sent to the administrations of the selected regional hospitals with a request for them to familiarize themselves with it and attract potential participants. In the informed consent form, the participants provided their data so that the researchers could contact them to distribute the questionnaire. As a result, the contact details of 400 nurses were collected. A link with an invitation to participate in the study was sent via WhatsApp messenger with a detailed description of the purpose of the study, the rights of the participants, and data protection. All the data received were anonymized and coded. The questionnaire was completely anonymous. The respondent had the right to refuse to participate in the study at any stage of data collection.

This study complied with all the principles of the Helsinki Declaration [15]. Ethical approval was obtained from the Local Commission on Bioethics of the West Kazakhstan Medical University named after M. Ospanov (Aktobe, Kazakhstan). The participants were informed about the purpose of the study and voluntarily took part in it. The participation was completely anonymous.

## 3. Results

In total, 284 nurses took part in the online survey. The average age of the respondents (Table 2) was 39.3 years (SD = 10.3); the average work experience was 16 years (SD = 10.37). A total of 33.8% (*n* = 96) of the respondents had diplomas of technical and professional education, 33.8% (*n* = 96) had graduated with applied bachelor’s degrees, 31.0% (*n* = 88) had higher education degrees, and 1.5% (*n* = 4) were master’s degree graduates.

Of the respondents, 77.8% (*n* = 221) held the position of hospital nurse; 6.3% (*n* = 18), advanced practice nurse; 15.9% (*n* = 45), nursing service manager; and 14.8% (*n* = 42), senior nurse. One-third of the respondents had worked in their position for 6–10 years. It was noted that work experience had an impact on the position held (Table 3), with a correlation coefficient of 0.364 (*p* = 0.01). Analysis determined a significant correlation between the age and work experience of the respondents: 0.772 (*p* = 0.01). This indicated a strong direct relation: the older the respondent, the longer their work experience.

The average values of the three burnout subscales were as follows: emotional exhaustion, 18.79 (SD = 11.71); depersonalization, 12.90 (SD = 7.57); personal accomplishment, 31.30 (SD = 15.29). These corresponded to the average levels of the formation of these indicators among nurses (*n* = 284) working in hospitals of the Republic of Kazakhstan. Moreover, 61.97% (EE ≥ 25 and/or DP ≥ 11) of the nurses had burnout out of the total number of observations. The distribution of respondents’ answers to questions regarding burnout is presented in Table 4. The most remarkable results are the following: 43% of the respondents on a daily basis provided more attention and care to others compared to the gratitude that they received from them in return. A total of 21.1% communicated with patients formally and without unnecessary emotions; 46.5% of the respondents could achieve a lot in their lives; 41.2% managed to do a lot in a day; 46.5% could easily communicate with patients and their relatives, regardless of their social status; 44.7% could create an atmosphere of friendliness and ease every day; 42.3% worked with pleasure and had plans for their professional careers; 36.6% noted that they could have a positive influence on patients’ mood on a daily basis; 42.3% were able to find the right solutions in difficult situations with patients and colleagues; 36.3% felt energetic and inspired at work; 46.1% understood the feelings of their patients well and used this understanding to successfully work with them; 45.1% never felt emotionally empty; 40.5% noted that they were never in a bad mood in the morning at the beginning of the working day; 37.7% never sought solitude; 50.4% did not experience any disappointments in life; 46.5% had never experienced a loss of interest in their professional duties.

Figure 1, Figure 2 and Figure 3 and Appendix A present the distributions of the development levels of the three main indicators of burnout (EE, DP, and PA) among hospital nurses in Kazakhstan depending on age, work experience, and position held. Thus, a high level of emotional exhaustion was found in 20 respondents aged 41–50 years, which is 13.74% of the total sample. Signs of depersonalization were shown by 59 respondents aged 41–50 years, which is 20.77% of the total number of participants. Also, a high level of depersonalization was found in 45 nurses aged 21–30 years, which is 15.85% of all the respondents. At the age of 31–40 years, a high level of depersonalization was revealed among 43 nurses, which is 15.14% of the total sample. A reduction in personal achievements was common among 37 nurses aged 41–50 years—13.03% of all the respondents—and among 31 nurses aged 21–30 years, which is 10.91% of the total number of study participants.

The rate of burnout was 72.46% among nursing specialists aged 21–30 years. The average values for the three burnout subscales were as follows: EE, 21.52 (SD = 13.096); DP, 14.74 (SD = 7.93); PA, 31.01 (SD = 15.01). In the age group of 31–40 years old, the burnout rate was 62.5%: EE, 14.60 (SD = 9.86); DP, 11.46 (SD = 7.29); PA, 30.89 (SD = 14.83). Among specialists aged 41–50 years, the burnout rate was 67.71%: EE, 21.55 (SD = 11.53); DP, 13.16 (SD = 7.61); PA, 31.32 (SD = 15. 83). In the age category of 51–60 years, the burnout rate was 59.57%: EE, 15.57 (SD = 10.00); de-DP, 11.91 (SD = 6.95); PA, 32.32 (SD = 15.70).

As a result of this study, it was determined that the highest level of EE occurred in northern and southern Kazakhstan (7.75%), as well as in central Kazakhstan—7.39% (Table 5). At the same time, high levels of DP were observed among nurses in Southern Kazakhstan (23.24%). High levels of PA were found in central Kazakhstan (13.03%).

It was noted that nursing specialists with work experience below 10 years had higher levels of EE (13.38%), DP (23.94%), and PA (15.49%). Similar prevalence was observed at the stage of the formation of EE (Figure 2).

The analysis of burnout among nurses occupying various positions determined that the highest levels of EE (22.54%), DP (50.00%), and PA (29.93%) were observed among ordinary nurses (Figure 3).

Thus, our study demonstrated that the level of burnout among hospital nurses in the Republic of Kazakhstan was fairly high and that burnout occurred in 61.97% of the respondents. Formed EE was present in almost one-third of the respondents (29.23%); DP, in 60.92%; and PA, in 38.73%. The formation stages according to these subscales occurred as follows: 25.7% (EE), 19.37% (DP), and 13.03% (PA). Signs of burnout are more common in the southern region of the Republic of Kazakhstan. Young nursing specialists are more susceptible to burnout than specialists possessing greater work experience.

## 4. Discussion

Burnout is an important psychological condition that affects nurses’ ability to work [3] and is characterized by emotional exhaustion, depersonalization, and reduced personal accomplishment [16]. EE is the main indicator of burnout. This is a feeling of fatigue and exhaustion from work. DP refers to negative attitudes toward patients and the mistreatment of people in the workplace. Reduced PA is a decrease in professional competence and achievements, which is caused by a decrease in the ability to fulfill one’s duties [17]. In conditions of constant communication with patients, their relatives, and colleagues, nurses work in a stressful environment [18].

Nurse burnout reduces quality of care and can put patients’ lives and health at risk [19]. A systematic review and meta-analysis of 85 studies found that nurse burnout is associated with poor quality of care, compromises patient safety, and contributes to decreased patient satisfaction with care [20].

Our study, conducted at the end of 2023, showed that 61.97% of nursing professionals suffer from signs of burnout. These data confirm the high occurrence of burnout among nurses. Thus, the results of the study conducted in July 2023 in China showed that 75.38% of nurses experienced symptoms of burnout [21]. A similar study conducted in Greece in 2021 determined that 71.6% of the participants at the time of the survey had signs of burnout [22]. Nurses aged 30–40 years were more likely to experience symptoms of burnout than others. These data somewhat differ from our research results. In this age category, burnout occurred in 62.5%, and at the age of 41–50 years, it occurred in 67.71%. The highest level of burnout was observed among young nursing professionals aged 21–30 years, at 72.46%.

In a study by Kazakh scientists conducted in Kazakhstan, the prevalence of high DP scores among doctors was 52% and that of high EE scores was 32% among doctors and 26% among nurses. This study showed no connection between burnout and age and length of service [9]. In 2020, before the onset of the COVID-19 epidemic, oncology health workers showed a higher burnout level: high EE, 47%; high DP, 63%; high RA, 59%. At the same time, there was no difference between doctors and nurses, with the exception of EE (42% vs. 59%, *p* < 0.01). The lower burnout scores were due to data collection in the pre-COVID-19 period. The results of this study are quite specific due to the oncology specialization of health workers, whose work is associated with severe patients [8].

Thus, our study showed that signs of burnout are more common among young (21–30 years old) and middle-aged (41–50 years old) nursing specialists. In Dyrbye’s work, a young age of nurses also predicted high levels of burnout [23].

In the work of Qedair [18], among 250 nurses working in the King Abdulaziz Medical City of National Guard Health Affairs (KAMC-JD) in Jeddah, no connection was found between burnout and the age of the respondents. It was noted that symptoms of burnout were more frequent particularly among hospital nurses (49%). Thus, in the USA, it was noted that work in hospitals was associated with a greater likelihood of developing burnout, resulting in the turnover of nursing staff, regardless of the performed functional duties. In 2018, 31.5% of nurses in the USA left their employment due to burnout. Additionally, 43.4% of nurses who were considering quitting their jobs identified burnout as one of the reasons contributing to their potential resignation [10].

Our study initially studied a sample of nurses working in hospitals in the Republic of Kazakhstan and showed that the level of DP of the respondents was 60.92%. In a study performed by Shahin [24], a high level of DP was found in 38% of nurses, while a high level of burnout was found in 39%.

Mangush’s study revealed the severe predominance of signs of a reduction in PA. According to the results of a study in hospitals, high-level EE occurred among 20% of nurses, and a moderate level occurred among 50%. DP occurred in 20% of cases, and PA in 70%. These data can be explained by poor self-assessment and the underestimation of professional achievements. As a result, symptoms of burnout cause the development of the worsening of relationships in the team [7].

A study conducted in Ethiopia in 2020 determined high levels of burnout among 207 nurses (56.5%). At the same time, EE was represented by the following data: 56.8%, high; 22.8%, moderate; 20.4%, low. DP had the following frequency of occurrence: 56.3%, high; 25.5%, moderate; 18.2%, low. PA presented as follows: 21.5%, high; 22.3%, moderate; 56.3%, low [25].

A systematic review of 113 studies from 49 countries conducted in 2020 found a pooled rate of 11.23% for nurse burnout symptoms worldwide. There are clear differences in the burnout prevalence depending on geographic location. The lowest prevalence of burnout was noted in Central Asia and Europe, and the highest was found in sub-Saharan Africa [26]. However, a tenth of nurses suffer from high levels of burnout, which requires intervention from the healthcare system. The WHO has identified burnout as an “occupational phenomenon” in the International Classification of Diseases (ICD-11). It is not a disease but a syndrome that occurs under long-term stress in the workplace [27]. Therefore, further study of burnout among nurses and the factors leading to it is important.

The results of Jean’s study [28] showed that environment and personal resources are the precursors of burnout within nurses. Thus, nurses who work by vocation experience less stress in the workplace and, as a result, are less susceptible to the symptoms of burnout. Therefore, it is important to maintain career aspirations among nurses by introducing effective strategies, reducing the workload in busy departments, paying attention to the restoration of emotional well-being in the workplace, and developing and maintaining nurse autonomy.

Thus, this study confirms the importance of burnout among nurses and, as a result, the need to prevent the development of burnout and its consequences to ensure safe nursing care for patients.

Strengths: Using a valid MBI-HSS-MP questionnaire. This study covered nurses providing round-the-clock nursing care in eight regional multidisciplinary clinical hospitals, which is 57% of the total number of such medical organizations in the Republic of Kazakhstan.

Weaknesses: The low readiness of nursing specialists in the Republic of Kazakhstan to actively participate in the study. This study did not consider the marital status of the respondents or the number of children in the family in the study of the development of burnout syndrome among nurses providing round-the-clock nursing care in a hospital setting.

## 5. Conclusions

During a study among nurses providing round-the-clock nursing care in multidisciplinary hospitals of the Republic of Kazakhstan, a high level of burnout (61.97%) was revealed within such indicators as emotional exhaustion (29.23%), depersonalization (60.92%), and personal accomplishment (38.73%). The level of burnout was higher among young nursing professionals, and burnout was more common in the southern regions of the Republic of Kazakhstan.

It is important to conduct a more detailed study of burnout among hospital nurses in the Republic of Kazakhstan to reduce potential risks to the health of nursing staff, ensure safe nursing care, improve the quality of nursing care, and reduce the outflow of personnel in the nursing specialty from practical healthcare.

## Figures and Tables

**Figure 1 nursrep-15-00092-f001:**
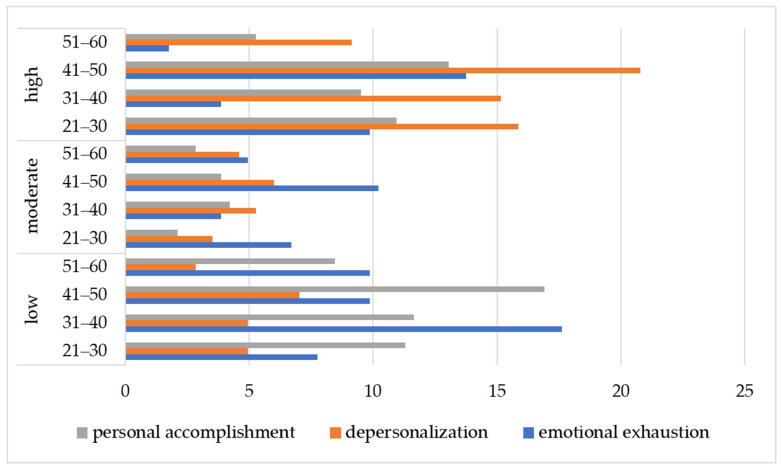
Prevalence of burnout levels among age categories of hospital nurses (*n* = 284).

**Figure 2 nursrep-15-00092-f002:**
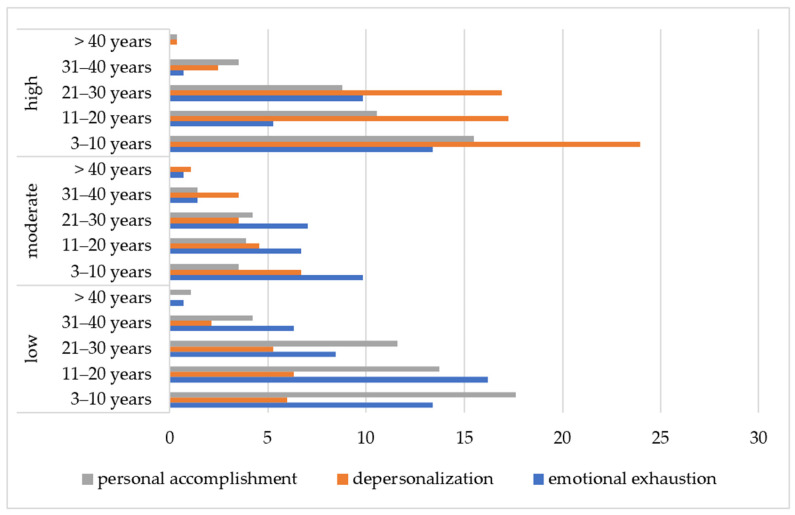
Prevalence of burnout levels depending on respondents’ work experience (*n* = 284).

**Figure 3 nursrep-15-00092-f003:**
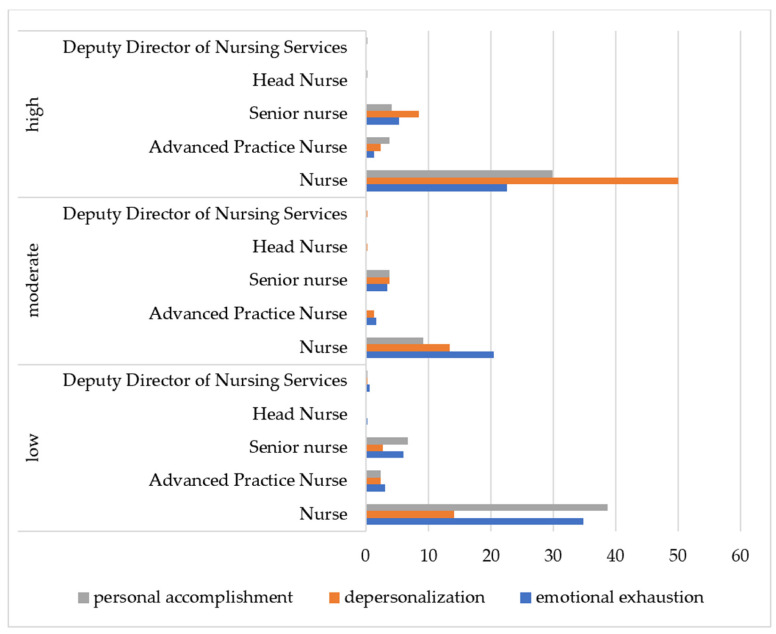
Prevalence of burnout levels among employees depending on the position they hold (*n* = 284).

**Table 1 nursrep-15-00092-t001:** Measuring scales of the burnout questionnaire for medical workers.

Subscale	Questions	Score
Emotional exhaustion	1, 2, 3, 6, 8, 13, 14, 16, 20	54
Depersonalization	5, 10, 11, 15, 22	30
Personal accomplishment	4, 7, 9, 12, 17, 18, 19, 21	48

**Table 2 nursrep-15-00092-t002:** Demographic data of the respondents (*n* = 284).

Index	Variables	Frequency	Percentage
Age	21–30	69	24.3
	31–40	72	25.4
	41–50	96	33.8
	51–60	47	16.5
Work experience	0–10 years	104	36.6
	11–20 years	80	28.2
	21–30 years	72	25.4
	31–40 years	24	8.5
	≥40 years	4	1.4
Years in position	0–5 years	61	21.5
	6–10 years	86	30.3
	11–15 years	51	18.0
	16–20 years	29	10.2
	21–25 years	29	10.2
	26–30 years	13	4.6
	31–35 years	10	3.5
	≥35 years	5	1.8

**Table 3 nursrep-15-00092-t003:** Correlation between respondents’ position and work experience (*n* = 284).

		Position	Age
Work experience	Pearson correlation	0.364 **	0.772 **
	Significance (bilateral)	0.000	0.000
	N	284	284

** The correlation is significant at the 0.01 level (two-tailed test).

**Table 4 nursrep-15-00092-t004:** Summary data for determining the level of burnout among hospital nurses in the Republic of Kazakhstan (*n* = 284).

Subscale	Item	Never	Seldom	More Rarely Than Often	Sometimes	More Often Than Rarely	Often	Daily
Emotional exhaustion	item 1	45.1	18.7	4.9	10.2	5.6	3.9	11.6
	item 2	24.6	16.5	16.2	10.6	4.9	8.1	19.1
	item 3	40.5	23.6	7.7	7.4	6.3	4.6	9.9
	item 6	9.8	12.0	10.2	8.1	14.1	9.5	36.3
	item 8	47.5	12.0	11.2	8.5	6.0	4.9	9.9
	item 13	50.4	16.5	7.4	11.3	4.9	1.8	7.7
	item 14	46.5	19.7	8.1	6.3	6.0	6.0	7.4
	item 16	37.7	14.1	11,9	11.3	3.9	6.7	14.4
	item 20	30.2	19.4	8.5	9.2	8.1	5.6	19.0
Depersonalization	item 5	29.2	16.9	9.9	6.7	9.9	6.3	21.1
	item 10	38.7	14.4	8.5	9.2	7.4	8.8	13.0
	item 11	38.7	14.1	11,9	9.2	9.9	2.1	14.1
	item 15	34.9	13.0	9.5	9.5	4.9	9.9	18.3
	item 22	14.1	11.3	6.7	6.3	8.7	9.9	43.0
Personal accomplishment	item 4	17.6	11.3	7.1	6.0	5.6	6.3	46.1
	item 7	9.2	12.3	7.0	9.9	11.6	7.7	42.3
	item 9	15.1	11.3	6.7	7.7	11.3	11.3	36.6
	item 12	11,9	10.6	7.4	10.2	7.7	9.9	42.3
	item 17	8.6	10.9	7.7	10.9	7.0	10.2	44.7
	item 18	11.6	11.6	7.0	6.0	4.6	12.7	46.5
	item 19	9.9	14.1	8.5	7.3	9.1	9.9	41.2
	item 21	8.8	11.6	9.2	8.1	7.0	8.8	46.5

**Table 5 nursrep-15-00092-t005:** Prevalence of development levels of three main indicators of burnout (EE, DP, and PA) among nurses in different regions of Kazakhstan (*n* = 284).

Level	Region of Kazakhstan	Emotional Exhaustion, *n* (%)	Depersonalization, *n* (%)	Personal Accomplishment, *n* (%)
Low	Western Kazakhstan	11 (3.87)	9 (3.17)	14 (4.93)
	Northern Kazakhstan	25 (8.80)	11 (3.87)	33 (11.62)
	Central Kazakhstan	27 (9.51)	15 (5.28)	24 (8.45)
	Southern Kazakhstan	65 (22.89)	21 (7.39)	66 (23.24)
Total		128 (45.07)	56 (19.72)	137 (48.24)
Moderate	Western Kazakhstan	2 (0.70)	0 (0.0)	7 (2.46)
	Northern Kazakhstan	19 (6.69)	17 (5.99)	6 (2.11)
	Central Kazakhstan	26 (9.15)	12 (4.23)	13 (4.58)
	Southern Kazakhstan	26 (9.15)	26 (9.15)	11 (3.87)
Total		73 (25.70)	55 (19.37)	37 (13.03)
High	Western Kazakhstan	18 (6.34)	22 (7.75)	10 (3.52)
	Northern Kazakhstan	22 (7.75)	38 (13.38)	27 (9.51)
	Central Kazakhstan	21 (7.39)	47 (16.55)	37 (13.03)
	Southern Kazakhstan	22 (7.75)	66 (23.24)	36 (12.68)
Total		83 (29.23)	173 (60.92)	110 (38.73)

## Data Availability

The data are contained within the article. The link for the data used in the article is as follows: https://drive.google.com/file/d/1i_BikKLBYfS1CLLk_llpM6s1c-rc2Zc_/view?usp=sharing (accessed on 3 March 2025).

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
