# Peer review of "Burnout Among Hospital Nurses in Kazakhstan"

_nursrep, 2025, doi:10.3390/nursrep15030092_

Round 1
Reviewer 1 Report
Comments and Suggestions for Authors
Thank you for inviting me to review this manuscript. The authors are to be congratulated on a well designed and executed cross-sectional survey. The results are clearly presented and the discussion relates the findings to the contemporary literature well. I was left with a couple of questions but these are probably for later papers, or further research. Could you connect these results to attrition rates in the different regions/contexts investigated? What do you think the differences between regions can be attributed to?
Author Response
Comments 1: Could you connect these results to attrition rates in the different regions/contexts investigated?
Response 1: Differences between regions may be due to different population densities. Thank you for pointing this out.
Comments 2: What do you think the differences between regions can be attributed to?
Response 2: The issue of personnel outflow is planned for further study.

Reviewer 2 Report
Comments and Suggestions for Authors
Dear Authors
It has been a pleasure to have the opportunity to review your manuscript “Burnout among hospital nurses in Kazakhstan”
I would like to make some recommendations to improve the paper:
ABSTRACT
Once the suggested changes have been made to the manuscript, consider redoing the abstract. (including the objective).
1. Introduction
This section should focus on the problem that has led to the development of this study. The increase in Burnout levels is not only associated with depersonalization, emotional exhaustion, and lack of personal accomplishment, all of which negatively impact the quality of care and may lead to professionals leaving their careers. It goes further, harming the professional’s health, who may present symptoms that could lead to illnesses. Physical symptoms include headaches, back pain, sleep disorders, nausea, muscle tension, and fatigue. From an emotional standpoint, individuals suffering from burnout may feel irritable, tense, and unmotivated. They may also feel indifferent, display cynicism, and have limited social interaction.
It is clear that to prevent this syndrome, healthcare institutions must engage in prevention efforts, aiming to detect it early and implement measures to impede its progression (psychological treatment, job reassignment, etc.).
In summary, it is not only necessary to define the syndrome but also to identify warning signs for early detection, emphasize the importance of prevention, outline the institution’s responsibility, and address treatment options. Ultimately, this study aims to identify the levels of the syndrome. For what purpose?
It is necessary to understand the situation in Kazakhstan—what do previous studies indicate?
LINES 47-48: The justification of the study needs to be improved, based on the points mentioned above.
LINES 49-51: The objective should be revised. It states, “The purpose of this study is to determine the syndrome of ‘burnout’ using three subscales: emotional exhaustion (EE), depersonalization (DP), and personal accomplishment (PA) among hospital nurses in the Republic of Kazakhstan.” However, in the RESULTS section, a correlation was performed to detect relationships between sociodemographic variables and subscales, which is not included in the stated objective.
2. Metodology
- LINES 53-92: The methodology does not meet the criteria for publication; it needs to be more rigorous and address many aspects that are absent in this document. This section is highly relevant in a research study as it forms the foundation of the work. It is recommended to organize it into more specific headings, such as; (e.g.,) setting, sampling & data collection, instrument or questionnaire, ethical considerations, and statistical analysis. Restructure this section accordingly.
- Since it is not structured, I will make general contributions for you to include in the corresponding subsections.
a. . Does this study adhere to the STROBE reporting guidelines for observational research? According to the "Guidelines and Standards Statement:..." (LINES 266-268), it does, but it actually does not, as there is a lack of a lot of information.
b. Period of study (date range)
c. What is the population under study and how large is it?
d. Type of sampling used?
e. Sample calculation indicating the minimum sample to be representative of the population? Sample calculation based on proportions or means?
f. Within the Republic of Kazakhstan, which health centres/hospitals were selected for distribution of the questionnaire? Please indicate the location of the centres by region as indicated in the RESULTS section.
g. How was the process of contacting potential participants who work in healthcare centres? Through management, head nurses, etc.? How many centres agreed to participate? How was the link to the digital questionnaire disseminated? Was there a prelude to the questionnaire containing “information for participants, data protection and request for explicit consent”? Was it anonymous?
h. LINE 87-92: Separate the content of this paragraph
“….was completely anonymous” refers to the process discussed above.
“…. The reliability (Cronbach's alpha) of the burnout questionnaire for medical workers (22 questions) was 0.910, and for the entire questionnaire it was 0.840 (33 questionnaire questions)..” refers to the instrument selected for the study, therefore its characteristics must be included in the subsection that refers to the instrument or questionnaire.
“The data was described using methods of descriptive and analytical statistics. The statistical package SPSS 23.0 (IBM SPSS Statistics, USA) was used to process the data. Significance criteria were calculated at p<0.05.” This content should be addressed in an exclusive section (eg. Statistical analysis) in addition to adding statistical relationship test used (in the RESULTS section) to relate two continuous variables.
i. I do not find the relevant authorization in the methodology (ethical aspects), please include that you have said authorization (can be found in the online manuscript 256-259)
3. Results
· It should include the number of participants and the percentage of participation as a result of the dissemination of the digital questionnaire (eg. If there are 8000 nurses in Kazakhstan, and they share the link to the questionnaire with everyone, but only 800 nurses participate, the participation was 10%)
· Table 5 can be simplified into a single row, since it is repetitive.
4. Discusion
LINES 188-189 “Our study, conducted at the end of 2023, showed that 66.20% of nursing professionals 188 suffer from signs of burnout” If we do not know the sample required to be representative (in methodology), I do not know if you can make this statement.
I think it is necessary to investigate the possible reasons for the increase in burnout levels in participating nurses from the southern part of the country. Is the economic, social, population situation... different from the rest of the country?
It does not include the limitations and strengths of the study, I suggest including them.
5. Conclusion
Although the “objective stated” provides an answer, it must go a little further, making contributions to improve the situation (prevention, involvement of the state or health centres, strategies to detect and avoid its increase…). As I mentioned in the introduction section, the objective lacks adding the relationships that have been found in the results section.
REFERENCE
I recommend carrying out a new search for evidence on the burnout situation in nursing, especially in Kazakhstan. There is an article from 2024 that is highly recommended to use: https://pubmed.ncbi.nlm.nih.gov/39441271/
I look forward to hearing from you and awaiting your replies and modifications.
Author Response
Comments 1: Once the suggested changes have been made to the manuscript, consider redoing the abstract. (including the objective).
Response 1: Thank you for pointing this out. We agree with this comment. Therefore, we have supplemented the abstract based on the results of the changes made to the article. The purpose of the study has been adjusted and included in the abstract lines 18-21. The methods part has also been adjusted: lines 22-23. Clarifications have been made in conclusions: lines 29-30.
Comments 2: This section should focus on the problem that has led to the development of this study. The increase in Burnout levels is not only associated with depersonalization, emotional exhaustion, and lack of personal accomplishment, all of which negatively impact the quality of care and may lead to professionals leaving their careers. It goes further, harming the professional’s health, who may present symptoms that could lead to illnesses. Physical symptoms include headaches, back pain, sleep disorders, nausea, muscle tension, and fatigue. From an emotional standpoint, individuals suffering from burnout may feel irritable, tense, and unmotivated. They may also feel indifferent, display cynicism, and have limited social interaction.
It is clear that to prevent this syndrome, healthcare institutions must engage in prevention efforts, aiming to detect it early and implement measures to impede its progression (psychological treatment, job reassignment, etc.).
In summary, it is not only necessary to define the syndrome but also to identify warning signs for early detection, emphasize the importance of prevention, outline the institution’s responsibility, and address treatment options. Ultimately, this study aims to identify the levels of the syndrome. For what purpose?
It is necessary to understand the situation in Kazakhstan—what do previous studies indicate?
LINES 47-48: The justification of the study needs to be improved, based on the points mentioned above.
LINES 49-51: The objective should be revised. It states, “The purpose of this study is to determine the syndrome of ‘burnout’ using three subscales: emotional exhaustion (EE), depersonalization (DP), and personal accomplishment (PA) among hospital nurses in the Republic of Kazakhstan.” However, in the RESULTS section, a correlation was performed to detect relationships between sociodemographic variables and subscales, which is not included in the stated objective.
Response 2: Agree. We have, accordingly, done/revised/changed/modified introduction to emphasize this point. Additions have been made to the introduction - page 1, paragraph 2, lines 42-43. Added data on burnout among medical workers in Kazakhstan – page 2, paragraph 1-2, lines 50-56. The purpose of the study has been adjusted – page 2, paragraph 3, lines 63-64.
Comments 3: LINES 53-92: The methodology does not meet the criteria for publication; it needs to be more rigorous and address many aspects that are absent in this document. This section is highly relevant in a research study as it forms the foundation of the work. It is recommended to organize it into more specific headings, such as; (e.g.,) setting, sampling & data collection, instrument or questionnaire, ethical considerations, and statistical analysis. Restructure this section accordingly.
- Since it is not structured, I will make general contributions for you to include in the corresponding subsections.
- . Does this study adhere to the STROBE reporting guidelines for observational research? According to the "Guidelines and Standards Statement:..." (LINES 266-268), it does, but it actually does not, as there is a lack of a lot of information.
- Period of study (date range)
- What is the population under study and how large is it?
- Type of sampling used?
- Sample calculation indicating the minimum sample to be representative of the population? Sample calculation based on proportions or means?
- Within the Republic of Kazakhstan, which health centres/hospitals were selected for distribution of the questionnaire? Please indicate the location of the centres by region as indicated in the RESULTS section.
- How was the process of contacting potential participants who work in healthcare centres? Through management, head nurses, etc.? How many centres agreed to participate? How was the link to the digital questionnaire disseminated? Was there a prelude to the questionnaire containing “information for participants, data protection and request for explicit consent”? Was it anonymous?
- LINE 87-92: Separate the content of this paragraph
“….was completely anonymous” refers to the process discussed above.
“…. The reliability (Cronbach's alpha) of the burnout questionnaire for medical workers (22 questions) was 0.910, and for the entire questionnaire it was 0.840 (33 questionnaire questions)..” refers to the instrument selected for the study, therefore its characteristics must be included in the subsection that refers to the instrument or questionnaire.
“The data was described using methods of descriptive and analytical statistics. The statistical package SPSS 23.0 (IBM SPSS Statistics, USA) was used to process the data. Significance criteria were calculated at p<0.05.” This content should be addressed in an exclusive section (eg. Statistical analysis) in addition to adding statistical relationship test used (in the RESULTS section) to relate two continuous variables.
I do not find the relevant authorization in the methodology (ethical aspects), please include that you have said authorization (can be found in the online manuscript 256-259)
Response 3: Thank you for pointing this out. We agree with this comment. Therefore, we have supplemented the methodology part. The methodology section is structured, the following subsections are highlighted: Study design and settings, Study participants, Data collection, Statistical analysis, Ethical consideration - page 2, lines 66-120.
These sections are also supplemented with missing information on the organization of the study and the formation of a representative sample population (page 2, lines 67-80). Exclusion and inclusion criteria have been added to the text (page 2, lines 82-85).
The ethical consideration section has been supplemented and the process of recruiting participants to the study is described in detail (page 2, lines 121-130).
Ethics committee approval is included in the supplement materials.
Comments 4: Results
- It should include the number of participants and the percentage of participation as a result of the dissemination of the digital questionnaire (eg. If there are 8000 nurses in Kazakhstan, and they share the link to the questionnaire with everyone, but only 800 nurses participate, the participation was 10%)
- Table 5 can be simplified into a single row, since it is repetitive.
Response 4: Thank you for pointing this out. We agree with this comment.
The methodology section presents data on sample formation. The total number of respondents is included in the results section – page 3, lines 137.
According to your recommendations, the data in the tables is repeated holidays. Table 3 contains the data of the correlation analysis – page 4, table 3 lines 152-153.
Comments 5: Discusion
LINES 188-189 “Our study, conducted at the end of 2023, showed that 66.20% of nursing professionals 188 suffer from signs of burnout” If we do not know the sample required to be representative (in methodology), I do not know if you can make this statement.
I think it is necessary to investigate the possible reasons for the increase in burnout levels in participating nurses from the southern part of the country. Is the economic, social, population situation... different from the rest of the country?
It does not include the limitations and strengths of the study, I suggest including them.
Response 5: Thank you for pointing this out. We agree with this comment.
The discussion section includes the strengths and limitations of the study – page 9, lines 239-247, 249-250.
Comments 6: Conclusion
Although the “objective stated” provides an answer, it must go a little further, making contributions to improve the situation (prevention, involvement of the state or health centres, strategies to detect and avoid its increase…). As I mentioned in the introduction section, the objective lacks adding the relationships that have been found in the results section.
Response 6: Thank you for pointing this out. We agree with this comment.
Expanded the conclusion based on literature search – page 10, lines 298-299, 306-307.

Reviewer 3 Report
Comments and Suggestions for Authors
Dear authors, thank you for sharing your work in this interesting field.
I have some comments that I hope you will find useful.
1. The objective of the report is to explore the phenomenon of burnout syndrome in hospital nurses in the Republic of Kazakhstan. It is necessary to provide context of the country's health system, at least about hospitals. Are they public or private hospitals? Is any of them a university hospital? In the materials and methods, there is no mention of the number of hospitals where the surveys were taken, although in the results, later in the paper, data is reported by region.
2. You established the exploration of burnout syndrome through the three subscales of the Maslach Burnout Inventory and provided information about the interpretation of each subscale. However, you do not explain how the levels of burnout are determined considering the high, moderate, and low scales that you later use in the results to evaluate age groups or the positions of the nurses.
3. It is relevant to establish the sample size and share with the reader the statistical assumptions used to determine the number of people you had to survey.
4. In the results, you make an evaluation between the position of the nurses and the years of experience; and between the age of the nurses and the years of experience. This does not appear later in the discussion. I suggest rethinking the relevance of these data.
5. The graphic content in Figure No. 1 cannot be adequately understood. If it remains the same, the reader cannot recognize which question the article number refers to, unless it has an annex or is included in the graphic.
6. The graphic content in Figure No. 2 cannot be adequately understood either. Three variables must be read together: burnout level (high, moderate, low), age group, and level reached in each MBI subscale. When observing it, it could be said that among those with a high level of burnout, those between 41 and 50 years old have the worst results in depersonalization compared to all the other groups. However, the volume of cases in each age group is unknown to understand if this is significant, or is a problem with the numbers of cases. There are only 284 respondents and if you divide it too much, each group will have few cases.
7. In Table 6, the results are set out by region. It has already been mentioned that the subject of how many hospitals, what characteristics, and what regions there are in the Republic of Kazakhstan should be included in the materials and methods. This is also a table that needs to be improved: It is not clear how many surveys come from which region. Again, a frequency table with three variables (burnout level, region, subscale) with a small sample runs the risk of generating tiny and insignificant groups.
8. There are additional variables in Burnout that are relevant: marital status and number of children. Without going any further, there is a meta-analysis “The effect of marital status on burnout levels of nurses: A meta-analysis Study published by Serkan Temel in the Journal of Clinical Medicine of Kazakhstan. If the data was collected, it is relevant to include it in the text. If they do not have it, they are incorporated in the discussion, as it constitutes a weakness of the article.
9. In the discussion, I noticed that the comparisons made are with China, Saudi Arabia, Ethiopia, and a global report. While you may have valid local arguments to compare with these countries, it would be wise to document those arguments clearly, especially given the global nature of the topic and the abundance of information available. If possible, please expand the review to provide more context.
10. The bibliography of the review could also be improved.
It is important for hospitals, centers, regions, or countries to understand the level of burnout present in organizations. This knowledge is a valuable contribution.Congratulations.
Author Response
Comments 1: The objective of the report is to explore the phenomenon of burnout syndrome in hospital nurses in the Republic of Kazakhstan. It is necessary to provide context of the country's health system, at least about hospitals. Are they public or private hospitals? Is any of them a university hospital? In the materials and methods, there is no mention of the number of hospitals where the surveys were taken, although in the results, later in the paper, data is reported by region
Response 1: Thank you for pointing this out. We agree with this comment. Therefore, we have added necessary information to methodology part – page 2, lines 70-72. The methodology section is structured, the following sections are highlighted: design and conditions of the study, study participants, data collection, statistical analysis, ethical considerations. Also, the data sections are supplemented with missing information on the study organization and the formation of a representative sample form. The participation of nurses from 8 regional multidisciplinary hospitals in the training is 57%.
Comments 2: You established the exploration of burnout syndrome through the three subscales of the Maslach Burnout Inventory and provided information about the interpretation of each subscale. However, you do not explain how the levels of burnout are determined considering the high, moderate, and low scales that you later use in the results to evaluate age groups or the positions of the nurses
Response 2: Agree. The Data collection section describes in detail the questionnaire and the method of analyzing the obtained data using the scales according to Vodop'yanova, a link to which is provided in the article. We added information about determining of burnout levels – page 3, lines 103, 105-106
Comments 3: It is relevant to establish the sample size and share with the reader the statistical assumptions used to determine the number of people you had to survey
Response 3: Agree. We added information about sampling in methodology part – page 2, lines 70-77. The study included eight regional multidisciplinary clinical hospitals (out of 14 in the country), which are geographically located in the regions of the Republic of Kazakhstan: Central, Northern, Southern, Western. The total number of nurses in these hospitals was 2032 employees, 52% of whom are involved in providing round-the-clock nursing care. According to the online calculator for calculating a representative sample for the general population, out of 1056 nurses providing round-the-clock nursing care in multidisciplinary hospitals the Sample size needed 282. This study covered 14% of nurses in regional hospitals included in the study
Comments 4: In the results, you make an evaluation between the position of the nurses and the years of experience; and between the age of the nurses and the years of experience. This does not appear later in the discussion. I suggest rethinking the relevance of these data
Response 4: Agree. According to your recommendation, data has been added to the discussion showing the influence of age and work experience on the level of emotional burnout among nurses. We added some discussion – page 9, lines 239-247, 249-250
Comments 5: The graphic content in Figure No. 1 cannot be adequately understood. If it remains the same, the reader cannot recognize which question the article number refers to, unless it has an annex or is included in the graphic
Response 5: Agree. According to your recommendation, Figure 1 had been supplemented with accompanying tables with data. Supplementary file – “Table to the Figure 1”
Comments 6: The graphic content in Figure No. 2 cannot be adequately understood either. Three variables must be read together: burnout level (high, moderate, low), age group, and level reached in each MBI subscale. When observing it, it could be said that among those with a high level of burnout, those between 41 and 50 years old have the worst results in depersonalization compared to all the other groups. However, the volume of cases in each age group is unknown to understand if this is significant, or is a problem with the numbers of cases. There are only 284 respondents and if you divide it too much, each group will have few cases.
Response 6: Agree. According to your recommendation, Figures 2-4 had been supplemented with accompanying tables with data. Supplementary file – “Table to the Figure 2-4”
Comments 7: In Table 6, the results are set out by region. It has already been mentioned that the subject of how many hospitals, what characteristics, and what regions there are in the Republic of Kazakhstan should be included in the materials and methods. This is also a table that needs to be improved: It is not clear how many surveys come from which region. Again, a frequency table with three variables (burnout level, region, subscale) with a small sample runs the risk of generating tiny and insignificant groups
Response 7: Thank you for pointing out it. Information about hospitals and regions is added to the methodology part – page 2, lines 70-77.
Comments 8: There are additional variables in Burnout that are relevant: marital status and number of children. Without going any further, there is a meta-analysis “The effect of marital status on burnout levels of nurses: A meta-analysis Study published by Serkan Temel in the Journal of Clinical Medicine of Kazakhstan. If the data was collected, it is relevant to include it in the text. If they do not have it, they are incorporated in the discussion, as it constitutes a weakness of the article.
Response 8: Thank you for pointing it out. Subsequent studies mention gradual changes that influence emotional burnout in Nurses.
Comments 9: In the discussion, I noticed that the comparisons made are with China, Saudi Arabia, Ethiopia, and a global report. While you may have valid local arguments to compare with these countries, it would be wise to document those arguments clearly, especially given the global nature of the topic and the abundance of information available. If possible, please expand the review to provide more context. The bibliography of the review could also be improved.
Response 9: Thank you for pointing it out. The introduction, discussion and bibliography of the article had been supplemented according to your recommendations.

Round 2
Reviewer 2 Report
Comments and Suggestions for Authors
Dear authors
I have carried out the Second round of revision of your manuscript “Burnout among hospital nurses in Kazakhstan””.
I recommend making minor revisions.
Next, I will make some recommendations to improve the document:
LINES 70-77. You must indicate the confidence level used and the margin of error to obtain a sample size of 283 participants.
RESULTS. Check the decimals, both in the text and in some tables (table 3), there are variations (e.g. SD 11.70 and 61.97%), put a period, not a comma.
DISCUSSION: you have included the suggestions made, however, it is necessary to address not only professional burnout but the consequences that these have on patient safety, such as decreased attention, increased irritability, low empathy, RISK OF ERRORS... You can include that studies justify the need for institutions and governments of countries to fight with strategies to avoid burnout and the consequences on patient safety.
IN CONCLUSIONS: include at the end the aspect of patient safety suggested in the discussion.
Good luck with your manuscript.
Author Response
Comments 1: LINES 70-77. You must indicate the confidence level used and the margin of error to obtain a sample size of 283 participants.
Response 1: Thank you for pointing this out. We agree with this comment. We added information regarding the reliability of the sample population. Lines 72-79, 82.
Comments 2: RESULTS. Check the decimals, both in the text and in some tables (table 3), there are variations (e.g. SD 11.70 and 61.97%), put a period, not a comma.
Response 2: Agree. Thank you for pointing this out. We corrected decimals according to your comments (text and tables)
Comments 3: DISCUSSION: you have included the suggestions made, however, it is necessary to address not only professional burnout but the consequences that these have on patient safety, such as decreased attention, increased irritability, low empathy, RISK OF ERRORS... You can include that studies justify the need for institutions and governments of countries to fight with strategies to avoid burnout and the consequences on patient safety.
Response 3: Thank you for pointing this out. We agree with this comment. Some data has been added to the discussion according to the received recommendation. Lines 240-243, 306-308.
Comments 4: IN CONCLUSIONS: include at the end the aspect of patient safety suggested in the discussion.
Response 4: Thank you for pointing this out. We agree with this comment. Finally, the patient safety aspect has been added. Lines 326-327.

Reviewer 3 Report
Comments and Suggestions for Authors
Dear Authors:
Thanks for replaying my observations. Some of them improve from the original manuscript, while others don´t get a change that would improve them.
To arrange Figure 1, you added a table in a supplemental item. I propose that you replace the graph with the table. It´s clearer and the lector can read numbers, especially for options "seldom" to "often" which are difficult to interpret on the graph. Also, you can order items by subscale. That can also help readers remember that emotional exhaustion and depersonalization will have opposite interpretations on a 0-6 frequency scale. Another option, one classic form to resume this type of scale is the use of media and st deviation to every question.
Figures 2, 3, and 4 must be clear about X scale. In my previous observation, I assumed that you were shown to readers the MBI subscale that was obtained in each subgroup. So, for example, in the 21-30-year-old group with low levels of burnout, they got 8 points in emotional exhaustion, 5 points in depersonalization, and 11 points in personal accomplishment (using the graph to identify the points, not the table). But, if we use the criteria to categorize the level of burnout (low EE 0-15 points, DP 0-5 points, and PA ≥37), none of the subgroups get three criteria. One explanation could be that you are using a media getting of each stratum. In that case, the problems could be with personal accomplishment results. In the figure and table that you offer in supplementary materials, the highest score is 13,03 in a 41-50-year-old group with a high level of burnout. In low levels burnout stratums, that supposedly must get more than 37 points, just get 16,9 points. So, there is no clear explanation to understand the numbers on a graph or table. Adding confusion, on the text you wrote that for all respondents between 21-30 years old, the results were EE 21.52 (SD 13.096), DP 14,74 (SD 7.03), and PA 31.01 (SD 15.01). That results are more consistent with the scale proposed by MBI ( EE 0 to 48 points, DP 0 o 30 points, and PA 0 to 48 points). So, you must be clear with data and consistent between text and graph/table to avoid confusing to readers.
Comment: One of the problems with MBI is that give us results for three aspects of burnout separately (https://hbr.org/2021/03/how-to-measure-burnout-accurately-and-ethically). It´s obvious that if people in one organization have negative results in three aspects, they have burnout. But there are several combinations possible. In a meta-analysis about "high level of burnout" in adult ICU, the authors found several ways to define "high level", a total score > -9, or several combinations of points of EE, DP, and PA. (https://pmc.ncbi.nlm.nih.gov/articles/PMC10041519/)
In table 4, please mention what do you mean by the number and number using in parenthenses. (person, percent, scale results, etc).
There is no mention of variables like marital status and number of children. I suggest mentioning the reason because you don´t take this variable and recognize it as a weakness to understand the Burnout phenomenon in Kazakhstan.
Good luck!!
Author Response
Comments 1: To arrange Figure 1, you added a table in a supplemental item. I propose that you replace the graph with the table. It´s clearer and the lector can read numbers, especially for options "seldom" to "often" which are difficult to interpret on the graph. Also, you can order items by subscale. That can also help readers remember that emotional exhaustion and depersonalization will have opposite interpretations on a 0-6 frequency scale. Another option, one classic form to resume this type of scale is the use of media and st deviation to every question.
Response 1: Thank you for pointing this out. We agree with this comment. According to your comments, Figure 1 has been replaced with a data table 4. Also, all statements are presented according to the MBI subscales. Page 5, table 4. Summary data on determining the level of burnout among hospital nurses in the Republic of Kazakhstan (n=284)
Comments 2: Figures 2, 3, and 4 must be clear about X scale. In my previous observation, I assumed that you were shown to readers the MBI subscale that was obtained in each subgroup. So, for example, in the 21-30-year-old group with low levels of burnout, they got 8 points in emotional exhaustion, 5 points in depersonalization, and 11 points in personal accomplishment (using the graph to identify the points, not the table). But, if we use the criteria to categorize the level of burnout (low EE 0-15 points, DP 0-5 points, and PA ≥37), none of the subgroups get three criteria. One explanation could be that you are using a media getting of each stratum. In that case, the problems could be with personal accomplishment results. In the figure and table that you offer in supplementary materials, the highest score is 13,03 in a 41-50-year-old group with a high level of burnout. In low levels burnout stratums, that supposedly must get more than 37 points, just get 16,9 points. So, there is no clear explanation to understand the numbers on a graph or table. Adding confusion, on the text you wrote that for all respondents between 21-30 years old, the results were EE 21.52 (SD 13.096), DP 14,74 (SD 7.03), and PA 31.01 (SD 15.01). That results are more consistent with the scale proposed by MBI ( EE 0 to 48 points, DP 0 o 30 points, and PA 0 to 48 points). So, you must be clear with data and consistent between text and graph/table to avoid confusing to readers.
Response 2: Thank you for pointing this out. We agree with this comment. In the results section, adjustments have been made to the presentation of data with clarifications in the figure titles. Figures 1-3 and the supplementary table present the distribution of development levels of the three main indicators of burnout (EE, DP, PA) among hospital nurses in Kazakhstan, depending on age, work experience and position. Line 183-193, pages 5-6. Figure 1-3, pages 6-8.
Comments 3: In table 4, please mention what do you mean by the number and number using in parenthenses. (person, percent, scale results, etc).
Response 3: Thank you for pointing this out. We agree with this comment. Table 4 has been renamed to Table 5 “Prevalence of levels of development of three main indicators of burnout (EE, DP, PA) among nurses in different regions of Kazakhstan” and indexes have been added. Table 4 on pages 6-7.
Comments 4: There is no mention of variables like marital status and number of children. I suggest mentioning the reason because you don´t take this variable and recognize it as a weakness to understand the Burnout phenomenon in Kazakhstan
Response 4: Your comment has been noted. This aspect is highlighted in the weaknesses of the study section. This study does not take into account the marital status of respondents and the number of children in the family due to the study of the development of burnout syndrome among nurses providing round-the-clock nursing care in a hospital setting. Lines 314-317, page 10.
